# Changes of Dissociative Properties of Hemoglobin in Patients with Chronic Kidney Disease

**DOI:** 10.3390/diagnostics14121219

**Published:** 2024-06-08

**Authors:** Justyna Korus, Maria Wydro, Maciej Gołębiowski, Kornelia Krakowska, Paweł Poznański, Kinga Musiał, Andrzej Konieczny, Hanna Augustyniak-Bartosik, Jakub Stojanowski, Mariusz Andrzej Kusztal, Tomasz Gołębiowski

**Affiliations:** 1Department of Nephrology and Transplantation Medicine, Wroclaw Medical University, Borowska 213, 50-556 Wroclaw, Poland; justyna.korus@student.umw.edu.pl (J.K.); wydro.maria@gmail.com (M.W.); maciej.golebiowski@student.umw.edu.pl (M.G.); korneliakrakowska006@gmail.com (K.K.); pawel.poznanski@umw.edu.pl (P.P.); andrzej.konieczny@umw.edu.pl (A.K.); hanna.augustyniak-bartosik@umw.edu.pl (H.A.-B.); jakub77xx@gmail.com (J.S.); mariusz.kusztal@umw.edu.pl (M.A.K.); 2Department of Pediatric Nephrology, Wroclaw Medical University, Borowska 213, 50-556 Wroclaw, Poland; kinga.musial@umw.edu.pl

**Keywords:** p50, metabolic acidosis, CKD

## Abstract

**Background**: The ability of hemoglobin to bind and dissociate oxygen is crucial in delivering oxygen to tissues and is influenced by a range of physiological states, compensatory mechanisms, and pathological conditions. This may be illustrated by the oxyhemoglobin dissociation curve (ODC). The key parameter for evaluating the oxygen affinity to hemoglobin is p50. The aim of this study was to evaluate the impact of hemodialysis on p50 in a group of patients with chronic kidney disease (CKD). An additional goal was to assess the correlation between p50 and the parameters of erythropoiesis, point-of-care testing (POCT), and other laboratory parameters. **Methods**: One hundred and eighty patients (106 male, 74 female), mean age 62.5 ± 17 years, with CKD stage G4 and G5 were enrolled in this cross-sectional study. Patients were divided into two groups, including 65 hemodialysis (HD) patients and 115 patients not receiving dialysis (non-HD). During the standard procedure of arteriovenous fistula creation, blood samples from the artery (A) and the vein (V) were taken for POCT. The causes of CKD, as well as demographic and comorbidity data, were obtained from medical records and direct interviews. **Results**: The weekly dose of erythropoietin was higher in HD patients than in non-HD patients (4914 ± 2253 UI vs. 403 ± 798 UI, *p* < 0.01), but hemoglobin levels did not differ between these groups. In the group of non-HD patients, more advanced metabolic acidosis (MA) was found, compared to the group with HD. In arterial and venosus blood samples, the non-HD group had significantly lower pH, pCO_2_ and HCO_3_^−^. This group had a higher proportion of individuals with MA with HCO_3_^−^ < 22 mmol/L (42% vs. 24%, *p* < 0.01). The absolute difference of p50 in arterial and venous blood was determined using the formula Δp50 = (p50-A) − (p50-V). Δp50 was significantly higher in the HD group in comparison to non-HD (0.08 ± 2.05 mmHg vs. −0.66 ± 1.93 mmHg, *p* = 0,02). There was a negative correlation between pH and the p50 value in arterial (pH-A vs. p50-A, r = −0.56, *p* < 0.01) and venous blood (pH-V vs. p50-V, r = −0.45, *p* < 0.01). In non-HD patients, hemoglobin levels correlated negatively with p50 (r = −0.29, *p* < 0.01), whereas no significant relation was found in HD patients. **Conclusions**: The ODC in pre-dialysis CKD (non-HD) patients is shifted to the right due to MA, and this is an additional factor influencing erythropoiesis. Hemodialysis restores the natural differences in hemoglobin’s dissociation characteristics in the arterial and venous circulation.

## 1. Introduction

The ability of hemoglobin to bind and dissociate oxygen is crucial for oxygen supplementation to tissues and may be affected by a variety of physiological conditions, compensatory mechanisms, and pathological conditions. The process may be illustrated by the oxyhemoglobin dissociation curve (ODC), which describes the relationship between the saturation of arterial blood hemoglobin with oxygen (SaO_2_) and the partial pressure of oxygen (PaO_2_) [1]. One of the phenomena influencing the ODC is the Bohr effect, a physiological phenomenon involving a reduction in the affinity of hemoglobin to bind oxygen in case of low pH and increased carbon dioxide partial pressure (pCO_2_) [2]. The reduced pH favorably affects the binding of H^+^ protons by hemoglobin and oxygen dissociation, reducing the affinity and facilitating oxygenation in the tissues [3]. The p50 is a key parameter for assessing the affinity of oxygen to hemoglobin which determines the partial pressure of the oxygen in the blood required to achieve 50% hemoglobin saturation. The p50 value varies depending on the source from which the blood was drawn. According to Balcerek et al. [4], the mean p50 value for capillary blood is 25.75 ± 0.72 mmHg, which corresponds to arterial blood [5], whereas the p50 value for venous blood ranges is 28.9 ± 1.25 mmHg for healthy women and 27.97 ± 1.54 for healthy men.

When the oxygen affinity of hemoglobin increases, there is a leftward shift in the ODC, which leads to a reduction in p50 [6]. Conversely, in a case where the affinity of hemoglobin for oxygen falls, its capacity to provide oxygen increases, the oxyhemoglobin dissociation curve shifts to the right, and p50 rises, as may be seen in an acidic environment with high pCO_2_ [7,8]. The p50 values in patients with chronic kidney disease (CKD) are higher than those in patients with normal kidney function, including patients with anemia [9]. The causes of these changes are phenomena occurring in conditions of chronic uremia, affecting p50 in different ways, frequently additive, but sometimes neutralizing each other. Anemia, organic phosphate accumulation, and a tendency for low pH and pCO_2_, as observed in metabolic acidosis (MA), are significant contributors to the rightward shift of the ODC. The decrease in pH and the associated increase in hydrogen ions facilitate proton binding by hemoglobin and subsequent oxygen dissociation [10]. The lower the pH, the easier the release of oxygen to the tissues. Thus, an increase in p50 in acidosis may protect tissues from organ dysfunction due to tissue hypoxemia, by increasing oxygen supply [11]. Regarding the body’s tissues, the ‘pH paradox’ refers to antiapoptotic, anti-inflammatory, and antioxidant mechanisms that provide cytoprotective effects. Acid pH impedes the penetration of calcium into cells, leading to slowing of its intracellular growth, which inhibits the action of enzymes activating degradative proteases, lipases, and nucleases, as well as causing cell necrosis. Acidosis inhibits apoptotic pathways and promotes the signaling of antiapoptotic mediators [12]. 

Patients with CKD develop hypoxia due to increased incidence of cardiovascular diseases, reducing the supply of oxygen to the tissues. Additional factors of tissue hypoxia include anemia, hyperglycemia, hypertension, smoking, hypercholesterolemia, and atherosclerosis [13]. Under hypoxic conditions, the elements contribute to the initiation of the apoptosis pathway, fibroblast and leukocyte activation, and subsequent extracellular matrix deposition (ECM). This eventually results in renal fibrosis and vessel narrowing, exacerbating hypoxia by limiting oxygen diffusion and reducing erythrocyte availability [14]. Hypoxia may be both a cause provoking, and a factor in, the progression of the disease [6]. As a result of progressive hypoxia, hypoxia-inducible factor (HIF-α) is triggered. This is a dimeric protein complex participating in the body’s response to hypoxia [15]. Its action is regulated by proteins containing the prolyl hydroxylase domain (PHD). HIF-1 is considered to be the crucial factor influencing erythropoietin (EPO) expression, as well as iron bioavailability. Hypoxia inhibits PHD activity and HIF-α accumulation. When it enters the cell nucleus as a transcription factor, it affects the expression of numerous target genes, including erythropoiesis [16].

Changes in the dissociation curve in CKD patients may impact the condition of tissue oxidation, altering the course of renal disease and the development of uremia complications such as anemia due to erythropoietin deficiency.

The main aim of this study was to evaluate changes in the p50 parameter in a group of patients with CKD, both in pre-dialysis and undergoing hemodialysis. 

The second goal was to assess the relationship between the p50 and parameters of erythropoiesis, point-of-care testing (POCT), and other laboratory parameters.

According to our knowledge, this is the first study that examines the dissociative properties of hemoglobin in CKD patients and the impact of hemodialysis on p50. 

## 2. Materials and Methods

Initially, 188 patients with CKD in stages G4 and G5, admitted from January 2022 to June 2023 for creating an arteriovenous fistula (AVF) for hemodialysis, were screened for eligibility for participation in the study. The inclusion criteria were: (1) age over 18 years; (2) CKD stage G4 and G5 qualified for AVF creation; (3) signed a written consent. Eight patients were excluded due to missing POCT samples. The remaining 180 patients, with a mean age 62.5 ± 17 years (117 males and 63 females), were enrolled in this cross-sectional study.

A collection of arterial and venous blood samples was conducted during the procedure of AVF creation. A longitudinal incision was made after the radial artery (RA) had been dissected and its distal and proximal sections clamped. Following the release of the clamp, blood was drawn into a heparinized syringe via a plastic needle inserted into the proximal part of the RA. The delay between RA closure and sample collection was no more than 30 s. A venous blood sample was aspirated after the vein was severed. The accompanying nurse then took the samples to the POCT in an adjacent room and promptly performed the test. The overall time from sample collection to measurement was no more than three minutes. An analyzer (RADIOMETER ABL90 SERIES, RADIOMETER MEDICAL APS, Brønshøj, Denmark) was utilized to investigate POCT parameters. This is a totally automatic piece of equipment that obtains diagnostic results while the patient is present or close by. It includes a cartridge of test reagents and performs an 8 h calibration. The following parameters were assessed: pO_2_, partial pressure of oxygen; sO_2_, oxygen saturation; HCO_3_^−^, bicarbonate; pCO_2_, carbon dioxide partial pressure; SBE, standard base excess; ABE, actual base excess; Na^+^, sodium; K^+^, potassium; Ca^2+^, ionized calcium; Cl^−^, chloride; AG, anion gap; and creatinine and urea. p50 is defined as the partial pressure of oxygen in blood at 50% oxygen saturation. It is an estimated parameter derived from pO_2_ and sO_2_ via extrapolation along the oxyhemoglobin dissociation curve (ODC) [17]. The differences in concentration between arterial (A) and venous (V) samples were presented as delta (Δ) and percentage of change (%). The mean p50 value was reported as the average of the p50-A and p50-V values, which theoretically correspond to the p50 value in the capillaries. The estimated glomerular filtration rate (eGFR) has been calculated using the CKD-EPI formula since the device did not automatically report it [18].

This group of 180 patients was divided into two subgroups: (1) non-dialysis patients with CKD (non-HD) and (2) CKD patients undergoing hemodialysis (HD). General clinical data are presented in Table 1.

The primary causes of CKD, as well as demographic and comorbidity details, were obtained from medical records and direct interviews. The Charlson Comorbidity Index (CCI) was counted using the rule described in the previous study [19]. In brief, this score encompasses the patient’s age and the following disorders reported in groups: (1) Heart diseases; chronic heart failure with exertional or paroxysmal nocturnal dyspnea and has responded to digitalis, diuretics, or afterload reducing agents; myocardial infarction (MI) with history of MI (EKG changes or enzyme changes); (2) Peripheral obstructive arterial disease (POAD) with intermittent claudication or past bypass for chronic arterial insufficiency, gangrene in the past or acute arterial insufficiency, or untreated thoracic or abdominal aneurysm (≥6 cm); (3) Cerebrovascular accident (CVA) or transient ischemic attacks (TIA) with minor or no residual deficit; dementia with chronic cognitive deficit; hemiplegia; (4) Chronic obstructive pulmonary disease (COPD); (5) Connective tissue disease; (6) Gastrointestinal complications: peptic ulcer disease, liver disease; (7) Diabetes mellitus; (8) Neoplasmatic disease: solid tumor; lymphoma; leukemia. According to the severity of the comorbidity, 1–6 points were allocated, and the sum was calculated.

In addition, ambulatory measurements of hemodynamic parameters were measured with a Mobil-O-Graph monitor (Industrielle Entwicklung Medizintechnik und Vertriebsgesellschaft GmbH–IEM, Stolberg, Germany), which records oscillometric arm blood pressure: systolic and diastolic blood pressure, pulse pressure, ejection fraction, cardiac output, and pulse waves. It determines the pulse wave velocity (PWV) as a measure of arterial stiffness. 

Statistical analysis was carried out using standard software (Statistica Version 13.3, StatSoft, Tulsa, OK, USA). Continuous variables between groups were reported as the mean and standard deviation (±SD) and compared using the dependent and independent *t*-test or Mann–Whitney U test based on the type groups compared and the normality of variables, tested using the Kolmogorov–Smirnov test. Categorical variables were reported as absolute (n) and percentage (%) and compared using the χ^2^ test. The relationship between POCT parameters was examined using Pearson’s correlation analysis. A *p*-value < 0.05 was considered significant.

Ethics approval was granted by the Ethics Board of Wroclaw Medical University, No. KB-609/2019.

## 3. Results

The study included 180 patients with CKD in stage G4 and G5, mean age 62.5 ± 17 years. The entire group was divided into two main subgroups, i.e., non-HD and HD patients. Non-HD and HD patients constituted 115 (63%) and 65 (36%) patients, respectively. The characteristics of the laboratory parameters are detailed in Table 2. There were no statistically significant differences in p50, with mean values of 27.01 ± 1.66 mmHg and 26.85 ± 1.63 mmHg for HD and non-HD, respectively. The non-HD group presented more advanced metabolic acidosis and more severe acid–base disorders in comparison to the HD group. In both arterial and venous blood samples, this subgroup had considerably lower pH, pCO_2_, and HCO_3_^−^ in comparison to the HD group. The anion gap (AG) was significantly higher in the HD group only in venous samples (V). Furthermore, the proportion of patients with metabolic acidosis, with HCO_3_^−^ concentrations less than 22 mmol/L, was higher in the non-HD subgroup. There were no significant differences in the values of p50-A and p50-V between the two subgroups; however, the absolute difference in p50 values between artery and vein (Δp50 and Δp50 (%)) was substantially higher in the HD group. In this group significantly higher CRP, alkaline phosphatase, lower urea, and lower total protein concentrations were found, in comparison to the non-HD group. HD patients administered a higher weekly dose of erythropoietin than non-HD patients (4914 ± 2253 UI vs. 403 ± 798 UI, *p* < 0.01). Although 6 (9%) HD patients and 89 (78%) non-HD patients did not receive erythropoietin, there was no significant difference in hemoglobin levels between these two groups. Our study used two different iron supplementation methods, oral supplementation for the non-HD and intravenous for the HD group. In the HD group, the mean weekly dose of intravenously injected iron(III)-hydroxide sucrose complex was 27.46 ± 27.66 mg. In the non-HD group, iron(III) hydroxide with polymaltose or iron(II) sulfate was used, and the mean daily dose of iron of was 42.46 ± 64.48 mg. HD patients received iron supplementation more frequently than non-HD (64% vs. 35%, *p* > 0.05). 

A significant negative relationship between pH and the p50 value has been found. Figure 1 shows this association in arterial blood, although a similar relationship was discovered in venous samples (pH-V vs. p50-V, r = −0.45, *p* < 0.01). The impact of carbon dioxide partial pressure (pCO_2_) on p50 was less extensive (pCO_2_-A vs. p50-A, r = −0.17, *p* < 0.05 and pCO_2_-V vs. p50-V, r = 0.14, *p* > 0.05). In the entire study group, there was no significant relationship between hemoglobin and p50-A, p50-V, and mean p50 (r = −0.018, r = −0.072, and r = −0.052, *p* > 0.05).

Following this, the POCT parameters of arterial and venous samples were compared (Table 3). In HD patients, normal differences in p50 value were observed, i.e., higher in venous samples (V) and lower in arterial (A), whereas in non-HD patients, there were no statistically significant differences between them. In all subgroups (A, V, non-HD, and HD), typical acid–base disorders (ABD) were found with MA. 

Following that, we conducted statistical analyses separately for the HD and non-HD subgroups. Mean p50 values were estimated for each subgroup, and POCT parameters were compared (Table 4). The mean p50 was 27.01 mmHg and 26.85 mmHg in the HD and non-HD group, respectively. According to these values, we divided patients into two additional subgroups, i.e., high and low p50 (p50 < 27.01 mmHg, p50 > 27.01 mmHg for HD patients and p50 < 26.85 mmHg, p50 > 26.85 mmHg for non-HD patients). In both subgroups with higher p50 value, the more advanced MA was observed with lower pH, HCO_3_^−^, ABE, and SBE. Acid–base disorders (ABD) showed the greatest severity in non-HD patients with a p50 value greater than 26.85 mmHg; this subgroup was identified by substantially elevated phosphate, while albumin, total protein, and hemoglobin levels were notably lower. The only significant difference in pCO_2_ between subgroups with a p50 < 26.85 mmHg and p50 > 26.85 mmHg was observed in the non-HD group. 

The mean p50 value was negatively correlated with the hemoglobin concentration in the group of non-HD patients (Figure 2). In contrast, no such correlation was found in the HD subgroup. 

## 4. Discussion

The main objective of this study was to assess the impact of CKD on the oxyhemoglobin dissociation curve (ODC). In our study, no significant p50 differences were observed between the HD and non-HD groups. 

The majority of scientific knowledge regarding the characteristics of the ODC in individuals with CKD is derived from physiological research carried out during the 1970s, which focused on patients with advanced CKD or those undergoing renal replacement therapy (RRT). In such patients, multiple biological variables, including typical CKD complications, acid–base balance disorders, and plasma phosphate retention, may each have an independent effect on the p50. Many patients with advanced CKD have metabolic acidosis (MA) [20], which may have a bivalent effect on the affinity of hemoglobin to oxygen. On the one hand, a low pH shifts the oxygen saturation of the hemoglobin’s dissociation curve to the right, known as the Bohr effect, resulting in improved oxygenation of the tissues. This finding was confirmed in our study, which showed a significant relationship between pH and p50 (see Figure 1) in the entire cohort of patients, both HD and non-HD. This was also confirmed by early studies by Bloomberg et al. prior to the implementation of erythropoietin, which additionally demonstrated a significant correlation between p50 and 2.3-BPG (r = 0.87, *p* < 0.001) in dialysis patients. Compared to healthy subjects, an increase in p50 (+52%, *p* < 0.001) was shown [21]. On the other hand, as studies by Rörth et al. [22] show, in the case of chronic acidosis there is a suppression of the production of 2.3-BPG, which in part neutralizes the Bohr effect, leading to almost constant affinity of oxygen at low pH values. The quoted study also found that increasing the concentration of inorganic phosphate stimulates intraerythrocytic synthesis of 2,3-diphosphoglycerinate, which moves the ODC to the right. Anemia itself in the population of patients with CKD also increases the production of 2.3-BPG. This is because low hemoglobin increases the concentration of deoxyhemoglobin, a stronger principle than oxyhemoglobin, and this intracellular alkalosis stimulates glycolysis and 2,3-BPG synthesis, resulting in an increase in p50 and better tissue oxidation [22].

According to our research, the dissociation features of hemoglobin oxygen in patients with CKD result from the influence of uremic-specific disturbances, such as acid–base disorders, serum inorganic phosphate retention, and anemia. Previous studies have shown that although p50 levels are elevated in patients with CKD and anemia, this increase is smaller compared to patients with anemia and normal kidney function [9].

In our study, p50 was evaluated in arterial blood (p50-A), venous blood (p50-V), and the mean of both values (mean p50), theoretically corresponding to blood in the capillaries. These parameters were evaluated respectively for non-HD and HD patients. In these two groups, p50 values did not differ significantly, and the highest and lowest values were observed in the HD group, in venous blood (p50-V = 26.68 ± 1.99 mmHg) and arterial blood (p50-A = 27.34 ± 1.66 mm Hg). (Table 1). When these values are compared to the typical p50 values in healthy individuals, whose mean p50 was 24.6 mmHg [20] in our study regardless of the group (HD or non-HD), there is a shift in the dissociation curve towards the right. The increase in p50 in patients with CKD is an adaptive mechanism and may have dissimilar practical significance in different clinical aspects. CKD is a disease correlating with the degree of interstitial fibrosis, renal tube destruction, the progression of numerous diseases, and kidney tissue hypoxia [23]. Typically, increased angiotensin II production is observed in CKD (ATII) [24]. It is formed as a result of a cascade of renin-dependent enzyme reactions and angiotensin-converting enzymes (ACE). Renin is produced by the cells of the juxtaglomerular apparatus of the kidneys, then acts on the angiotensinogen (produced in the liver), resulting in the formation of angiotensin I (ATI). Its conversion into ATII is influenced by increased synthesis of renin, which is observed in patients with CKD. Recent study has shown that ATII induces the formation of 2.3-BPG in erythrocytes via the enzyme sphingomyelin kinase, which reduces hemoglobin’s affinity for oxygen and hypoxic effects, potentially slowing the progression of CKD [6]. 

The second issue raised by the increase in p50 is its effect on erythropoiesis and the need to correct hemoglobin concentrations in CKD patients. In the current study, patients with HD required significantly higher doses of EPO to maintain similar hemoglobin values, compared to those without HD (4914 ± 2253 UI vs. 403 ± 798 UI, *p* < 0.01). While the reasons for these differences may, of course, be related to various factors, the most crucial is the greater impairment of erythropoietin (EPO) production in more advanced kidney disease in HD patients. In Table 4, it can be concluded that hemoglobin values were substantially higher only in the non-HD group in patients with lower p50, i.e., the mean p50 < 26.85 mmHg subgroup, which was characterized by less advanced acidification, with higher values of pH, HCO_3_^−^, and pCO_2_. In addition, in the non-HD group, hemoglobin levels were negatively correlated with mean p50 (Figure 2). We assumed that in this group (non-HD) the relative shift of the dissociation curve to the left (compared to the HD group) is the result of less advanced acidosis. As a result, in this group, less oxygen is delivered to oxygen-sensitive tissues, including the kidneys, which stimulates increased production of EPO in tissue hypoxia conditions via the HIF pathway, contributing to the maintenance of erythropoiesis and less demand for exogenous EPO, when compared to the HD group. It appears that changes in the dissociation curve in CKD can be compared to the hereditary hemoglobin variant with abnormally low oxygen affinity, which is a rare cause of anemia and/or cyanosis [25]. The affected individuals release oxygen to the tissue much more efficiently than usual, resulting in a decreased stimulation of erythropoietin production from the kidneys, which leads to a reduction in erythrocyte formation and anemia. These variants of low-affinity hemoglobin are associated with increased p50. In addition, it should be noted that dialyzed patients, despite significant anemia, often do not report general symptoms, which is the result, as Hertig and Ferrer-Marin [26] elegantly described, of a trend of a rightward shift of the dissociation curve in HD patients. In this study, increased production of 2.3-DPG, direct urea-induced modifications in the structure of hemoglobin, isocyanic acid, and long dialysis period are directly responsible for high p50 in these patients. Consequently, balanced anemia alignment is proposed by the authors, as adaptive modifications in the dissociation curve guarantee sufficient oxygen delivery notwithstanding reduced hemoglobin levels.

Physiologically, in arterial samples p50 is significantly lower than in venous samples. In animal experiments (beagle dogs), arterial and venous blood p50 values recorded by the blood gas analyzer were 28.21 ± 1.32 mmHg and 37.60 ± 4.80 mmHg, respectively [27]. In horses, the values were 25.5 ± 2.1 mmHg (arterial blood) and 26.1 ± 2.1 mmHg (venous blood) [28]. In the current study, when comparing non-HD and HD patients, no difference was observed between the individual values of the p50 parameter, taken from both the artery (p50-A) and the vein (p50-V) (Table 1). Significant intergroup variations were identified in the absolute and percentage p50 discrepancies between arterial and venous samples (Δp50 and Δp50%), which were 0.08 mmHg and −0.01%, respectively, in the non-HD group and −0.66 mmHg and −2.78% in the HD group (Table 1). These data indicate that hemodialysis restores the physiological differences in the oxidative properties of hemoglobin, i.e., significantly reduces this p50 value in arterial blood and at the same time increases this value in venous blood. To accurately analyze the dependencies in Table 3, POCT parameters are compared separately for arterial and venous blood. The p50-A and p50-V values in HD patients differ significantly (26.68 ± 1.99 vs. 27.34 ± 1.84, *p* < 0.01). In the non-HD group there were no significant differences (26.89 ± 2.02 vs. 26.81 ± 1.83, *p* = 0.66), with nearly comparable values. The rigid characteristics of p50 values may serve as an indicator of advanced ABD disorders in pre-dialysis patients. All parameters describing these ABD disorders i.e., pH-A, pH-V, p-CO_2_-A, p-CO_2_-V, HCO_3_^−^-A, HCO_3_^−^-V, BE-A, and BE-V, reflect deeper acidosis in the non-HD group, which may indicate that in this group we are dealing with a deep depletion of the bicarbonate buffer, which contributes to the disappearance of physiological differences between the arterial and venous systems. This phenomenon has been described in our previous work [29].

Summarizing our findings, we should also consider the study’s limitations. Anemia in chronic renal disease patients can be caused by many factors, including low eGFR, diabetes, high blood phosphate levels, proteinuria, and other conditions [30]. Not all predictors were deeply investigated in this study, including EPO dose and iron supplementation therapy. First, we observed a high standard deviation (SD) for EPO dose in the results. High SD mostly affected non-HD patients and results from the fact that not all patients received EPO during predialysis, and their dosage was tailored to the recommended Hb level of 11 g/dL [31]. The study examined non-HD and HD populations without evaluating a wide range of inflammatory markers, whereas none of the patients presented with active infection at the time of AVF creation. We did not estimate hepcidin level, which regulates iron metabolism by inhibiting intestinal iron absorption and iron release from iron stores and is the marker of inflammation. The DOPPS study showed that a higher Hb level above 11 g/dL is associated with a lower ferritin level in dialysis patients [32]. This inverse relationship may be related to hidden inflammation, e.g., in patients with catheters or non-functioning vascular grafts, as described by Nassar et al. [33]. In these situations, a high level of ferritin may be a marker of inflammation. In our study, we observed significantly higher CRP levels in dialysis patients compared to the non-HD group, which may affect erythropoiesis, while ferritin levels were comparable. 

Iron supplementation plays a pivotal role in anemia correction. Many studies indicate that an optimum iron-deficit therapy regimen may achieve and maintain Hb levels quickly, and delay or eliminate the requirement of additional anemia therapies, including erythropoiesis-stimulating agents [34]. Our study used two different iron supplementation methods, oral supplementation for the non-HD and intravenous for the HD group. Although our analysis showed that dialysis patients were more likely to receive iron supplementation, comparing the HD and non-HD groups in this respect is difficult and may also affect the correction of anemia.

Due to the cross-sectional nature of the study, we were not able to evaluate patients for p50 changes in the context of CKD progression, nor did we examine other factors that could potentially influence the characteristics of the ODC, e.g., 2,3-BPG. Moreover, this was a single-center study with a relatively small number of patients included. The observed correlations between hemoglobin and mean p50 were not strong and may only suggest that changes in the dissociation curve may be one of the factors responsible for erythropoiesis. However, we are confident that the findings accurately reflect changes in hemoglobin’s dissociative characteristics in individuals with CKD, as well as alterations that occur at the beginning of renal replacement therapy.

## 5. Conclusions

ODC in patients with CKD (non-HD) before dialysis is shifted to the right due to their acid–base abnormalities. Hemodialysis treatment restores the natural differences in the characteristics of hemoglobin dissociation in the arterial and venous circulation. We assume that this is an additional factor affecting erythropoiesis.

## Figures and Tables

**Figure 1 diagnostics-14-01219-f001:**
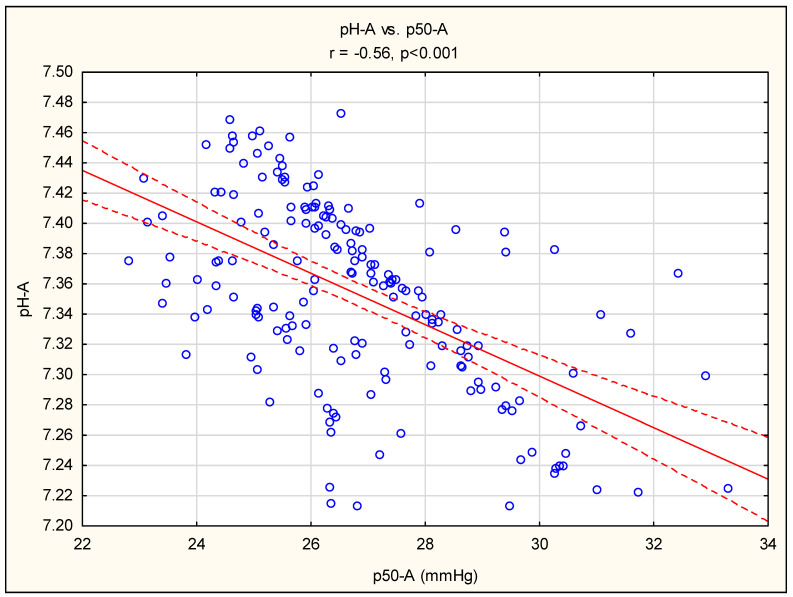
Pearson’s correlation between pH-A and p50A. Abbreviation: A, arterial sample. The red line describes the linear correlation line. The red-dashed line is the standard deviation line. A blue circle represents each case of study.

**Figure 2 diagnostics-14-01219-f002:**
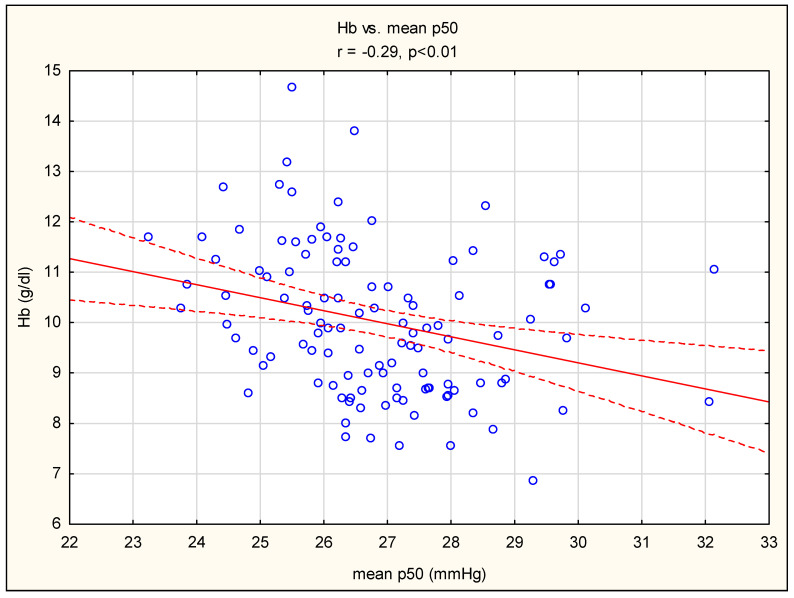
Pearson’s correlation between Hb and mean p50 in non-HD. The red line describes the linear correlation line. The red-dashed line is the standard deviation line. A blue circle represents each case of study.

**Table 1 diagnostics-14-01219-t001:** The study group’s overall clinical features divided into two main subgroups.

Causes of CKD				
	non-HD, *N* = 115	HD, *N* = 65	*p* *	No (%)
DM and HA (%)	55 (48)	18 (28)	>0.05	86 (48)
Chronic GN (%)	31 (27)	18 (28)	>0.05	49 (27)
ADPKD (%)	10 (9)	2 (0)	>0.05	12 (6)
IN (%)	5 (4)	4 (3)	>0.05	9 (5)
Others (%)	12 (10)	12 (18)	>0.05	24 (13)
Comorbidities **				
	non-HD, *N* = 115	HD, *N* = 65	*p*	All study patients *N* = 180
CCI (points)	6.46 (3.05)	6.52 (3.21)	>0.05	6.48 (3.09)
	non-HD *N* = 115	HD, *N* = 65	*p* *	No (%)
Heart diseases (%)	74 (64)	54 (83)	>0.05	128 (71)
Peripheral vascular disease (%)	27 (23)	16 (25)	>0.05	43 (23)
Cerebrovascular accident (%)	34 (39)	21 (32)	>0.05	55 (31)
Chronic obstructive pulmonary disease (%)	10 (9)	6 (9)	>0.05	16 (8)
Connective tissue disease (%)	12 (10)	2 (3)	>0.05	14 (8)
Gastrointerstitial complications (%)	16 (14)	6 (9)	>0.05	32 (18)
Diabetes mellitus (%)	39 (34)	24 (37)	>0.05	63 (35)
Neoplasmatic disease (%)	15 (13)	10 (15)	>0.05	25 (14)

Student’s *t*-test for independent variables, * chi quadrat test, *p*-value < 0.05 statistically significant. Abbreviations: DM, diabetes mellitus; GN, glomerulonephritis; IN, interstitial nephritis; CCI, Charlson Comorbidity Index; CKD, chronic kidney disease; HA, hypertension; ADPKD, autosomal dominant polycystic kidney disease; ** Comorbidities were included in groups in the Materials and Methods section.

**Table 2 diagnostics-14-01219-t002:** The study group’s laboratory characteristics divided into two subgroups.

	non-HD, *N* = 115	HD, *N* = 65	*p*
pH-A	7.33 ± 0.06	7.38 ± 0.05	<0.01
pCO_2_-A (mmHg)	34.44 ± 4.98	37.06 ± 4.66	<0.01
pO_2_-A (mmHg)	85.64 ± 16.33	82.71 ± 16.89	>0.05
HCO_3_^−^-A (mmol/L)	18.49 ± 3.99	22.70 ± 2.95	<0.01
ABE-A (mmol/L)	−6.62 ± 4.34	−2.07 ± 3.64	<0.01
sO_2_-A (%)	95.81 ± 3.11	95.22 ± 2.97	>0.05
K^+^-A (mmol/L)	4.41 ± 0.78	4.77 ± 0.75	<0.01
Na^+^-A (mmol/L)	140.56 ± 3.17	139.06 ± 3.06	<0.01
Ca^2+^-A (mmol/L)	1.14 ± 0.10	1.09 ± 0.09	<0.05
Cl^−^-A (mg/dL)	112.31 ± 5.55	105.78 ± 4.17	<0.01
Anion Gap-A	9.78 ± 2.33	10.49 ± 2.34	>0.05
pH-V (mmHg)	7.27 ± 0.07	7.32 ± 0.06	<0.01
pCO_2_-V (mmHg)	37.06 ± 5.04	41.09 ± 4.65	<0.01
pO_2_-V (%)	60.53 ± 13.30	56.20 ± 11.11	<0.05
HCO_3_^−^-V (mmol/L)	17.62 ± 4.02	20.68 ± 2.93	<0.01
ABE-V (mmol/L)	−8.47 ± 4.67	−4.41 ± 3.79	<0.01
sO_2_-V (%)	88.56 ± 7.03	86.10 ± 8.00	<0.05
K^+^-V (mmol/L)	4.47 ± 0.75	4.89 ± 0.68	<0.01
Na^+^-V (mmol/L)	141.06 ± 3.03	139.55 ± 2.81	<0.01
Ca^2+^-V (mmol/L)	1.17 ± 0.11	1.12 ± 0.09	<0.05
Cl^−^-V (mg/dL)	112.34 ± 5.40	105.63 ± 3.93	<0.01
Anion gap-V (mmol/L)	11.17 ± 2.49	12.39 ± 2.37	<0.01
ΔpCO_2_ (mmHg)	−2.69 ± 2.67	−4.10 ± 3.74	<0.01
ΔcHCO_3_^−^ (mmol/L)	0.87 ± 1.79	1.13 ± 2.41	>0.05
Total protein (g/dL)	5.95 ± 0.93	6.30 ± 0.80	<0.05
Albumin (g/dL)	3.37 ± 0.59	3.49 ± 0.48	>0.05
TC (mg/dL)	192.11 ± 67.62	179.10 ± 55.73	>0.05
TG (mg/dL)	156.91 ± 81.20	141.35 ± 93.61	>0.05
CRP (mg/L)	8.78 ± 14.5	14.43 ± 19.9	<0.05
Pi (mg/dL)	5.48 ± 1.27	5.59 ± 1.59	>0.05
PTH (pg/mL)	350.01 ± 229.76	412.89 ± 348.28	>0.05
AP (UI/L)	75.09 ± 28.83	91.52 ± 66.71	<0.05
BNP (pg/mL)	563.10 ± 771.42	2231.29 ± 5930.58	<0.05
sCr (mg/dL)	5.38 ± 1.83	6.64 ± 2.44	<0.01
eGFR (mL/min/1.75 m^2^)	11.67 ± 4.24	10.01 ± 4.54	<0.05
Urea (mg/dL)	146.67 ± 38.55	103.34 ± 35.42	<0.01
p50-A (mmHg)	26.89 ± 2.02	26.68 ± 1.99	>0.05
p50-V (mmHg)	26.80 ± 1.83	27.34 ± 1.84	>0.05
Δp50 (mmHg)	0.08 ± 2.05	−0.66 ± 1.93	<0.05
Δp50 (%)	−0.01 ± 7.44	−2.77 ± 7.31	<0.05
Mean p50 (mmHg)	26.85 ± 1.63	27.01 ± 1.66	>0.05
Parameters of erythropoiesis			
Hb (g/dL)	10.02 ± 1.46	10.29 ± 1.78	>0.05
Fe (μg/dL)	62.09 ± 32.97	61.79 ± 29.41	>0.05
TIBC (μg/dL)	222.34 ± 57.43	231.26 ± 57.36	>0.05
TSAT (%)	28.29 ± 14.52	27.78 ± 13.64	>0.05
Ferritin (ng/mL)	238.94 ± 245.46	274.31 ± 186.4	>0.05
Weekly dose of erytropoetin (U/week)	403.51 ± 798.65	4914.89 ± 2253.79	<0.01
Iron supplementation (%)	40 (35)	41 (64)	<0.05 *
Clinical parameters			
PWV (m/s)	10.11 ± 2.38	9.64 ± 2.59	>0.05
CCI (point)	6.46 ± 3.05	6.52 ± 3.21	>0.05
Smoking (pack-year)	9.93 ± 15.19	14.01 ± 14.56	>0.05
The type of metabolic acidosis			
	non-HD, *N* = 115	HD, *N* = 65	*p* *
pH-A < 7.33 (%)	52 (45)	11 (17)	<0.01
Non-MA-A (%)	24 (21)	42 (62)	<0.01
AGMA-A (%)	49 (43)	8 (12)	<0.01
HAGMA-A (%)	42 (37)	15 (23)	>0.05
Anion gap-A ≥ 10mmo/L (%)	53 (46)	35 (54)	>0.05
HCO_3_^−^-A ≤ 22 mmol/L (%)	91 (79)	23 (35)	<0.01
Non-MA-V (%)	16 (14)	22 (34)	<0.05
AGMA-V (%)	28 (24)	0 (0)	<0.01
HAGMA-V (%)	71 (62)	43 (66)	>0.05
Anion gap V ≥ 10 mmol/L (%)	82 (71)	65 (100)	>0.05
HCO_3_^−^-V ≤ 22 mmol/L (%)	91 (86)	43 (66)	>0.05

Student’s *t*-test for independent variables, * chi quadrat test, *p*-value < 0.05 indicates statistical significance. Abbreviations: HAGMA, high anion gap metabolic acidosis; AGMA, normal anion gap metabolic acidosis; non-MA, non-metabolic acidosis; SD, standard deviations; pO_2_, oxygen partial pressure; pCO_2_, carbon dioxide partial pressure; HCO_3_^−^, bicarbonate; ABE, actual base excess; Na^+^, sodium; K^+^, potassium; Cl^−^, chloride; Ca^2+^, ionized calcium; AG, anion gap; TSH, thyroid stimulating hormone; TC, total cholesterol; TG, triglycerides; Pi, phosphate; PTH, parathormone; AP, alkaline phosphatase; CRP, C-reactive protein; BNP, B-type natriuretic peptide; eGFR, estimated glomerular filtration rate; Hb, hemoglobin, Fe, iron, TIBC, total iron-binding capacity, TSAT, transferrin saturation, PWV, pulse wave velocity; CCI, Charlson Comorbidity Index; A, arterial sample; V, venous sample.

**Table 3 diagnostics-14-01219-t003:** Comparison of selected POCT parameters in arterial and venous samples, presented separately for non-HD and HD patients.

**non-HD**	**A**	**V**	** *p* **
p50 (mmHg)	26.89 ± 2.02	26.81 ± 1.83	0.66
pH	7.33 ± 0.06	7.27 ± 0.07	<0.01
pCO_2_ (mmHg)	34.44 ± 4.98	37.06 ± 5.04	<0.01
HCO_3_^−^ (mmol/L)	18.49 ± 3.99	17.62 ± 4.02	<0.01
Anion gap (mmol/L)	9.78 ± 2.33	11.17 ± 2.49	<0.01
ABE (mmol/L)	−6.62 ± 4.34	−8.47 ± 4.67	<0.01
SBE (mmol/L)	−7.37 ± 4.78	−9.14 ± 5.06	<0.01
**HD**	**A**	**V**	** *p* **
p50 (mmHg)	26.68 ± 1.99	27.34 ± 1.84	<0.01
pH	7.38 ± 0.05	7.32 ± 0.06	<0.01
pCO_2_ (mmHg)	37.06 ± 4.66	41.09 ± 4.65	<0.01
HCO_3_^−^ (mmol/L)	22.70 ± 2.95	20.68 ± 2.93	<0.01
Anion gap (mmol/L)	10.49 ± 2.34	12.39 ± 2.37	<0.01
ABE (mmol/L)	−2.07 ± 3.64	−4.41 ± 3.79	<0.01
SBE (mmol/L)	−2.39 ± 4.00	−4.61 ± 4.07	<0.01

Student’s *t*-test for dependent variables, *p*-value < 0.05 statistically significant. Abbreviation: A, arterial sample; V, vein sample; HD, hemodialysis; pCO_2_, carbon dioxide partial pressure; HCO_3_^−^, bicarbonate; SBE, standard base excess; ABE, actual base excess.

**Table 4 diagnostics-14-01219-t004:** Comparison of POCT parameters in HD and non-HD subgroups based on mean p50 values.

	HD			non-HD		
	p50 < 27.01 mmHg, *N* = 38	p50 > 27.01 mmHg, *N* = 27	*p*	p50 < 26.85 mmHg, *N* = 63	p50 > 26.85 mmHg, *N* = 52	*p*
pH-A	7.41 ± 0.04	7.35 ± 0.05	<0.01	7.35 ± 0.04	7.30 ± 0.06	<0.01
pCO_2_-A (mmHg)	37.07 ± 5.01	37.05 ± 4.17	0.99	35.34 ± 4.20	33.36 ± 5.64	<0.05
pO_2_-A (mmHg)	83.10 ± 17.60	82.13 ± 16.12	0.82	82.66 ± 12.95	89.25 ± 19.19	<0.05
HCO_3_^−^-A (mmo/L)	23.79 ± 2.70	21.16 ± 2.64	<0.01	20.03 ± 3.16	16.63 ± 4.12	<0.01
ABE-A (mmo/L)	−0.73 ± 3.26	−3.95 ± 3.36	<0.01	−4.80 ± 3.26	−8.84 ± 4.49	<0.01
SBE-A (mmo/L)	−0.96 ± 3.62	−4.41 ± 3.67	<0.01	−5.43 ± 3.65	−9.72 ± 4.96	<0.01
sO_2_-A (%)	95.80 ± 2.67	94.41 ± 3.22	0.06	96.11 ± 1.71	95.45 ± 4.23	0.26
K^+^-A (mmo/L)	4.62 ± 0.73	4.97 ± 0.74	0.06	4.37 ± 0.72	4.45 ± 0.85	0.56
Na^+^-A (mmo/L)	138.97 ± 3.20	139.18 ± 2.90	0.79	140.38 ± 3.15	140.78 ± 3.21	0.50
Ca^2+^-A (mmo/L)	1.10 ± 0.10	1.08 ± 0.09	0.59	1.15 ± 0.11	1.12 ± 0.10	0.23
Cl^−^-A (mg/dL)	105.00 ± 4.03	106.85 ± 4.19	0.08	110.61 ± 4.66	114.36 ± 5.89	<0.01
AG-A (mmo/L)	9.87 ± 2.20	11.35 ± 2.30	0.01	9.74 ± 2.15	9.82 ± 2.55	0.84
pH-V	7.35 ± 0.05	7.27 ± 0.05	<0.01	7.30 ± 0.06	7.24 ± 0.07	<0.01
pCO_2_-V (mmHg)	40.67 ± 3.82	41.69 ± 5.67	0.39	37.90 ± 4.47	36.04 ± 5.54	<0.05
pO_2_-V (mmHg)	56.82 ± 11.16	55.29 ± 11.20	0.59	58.75 ± 10.93	62.65 ± 15.50	0.12
HCO_3_^−^-V (mmo/L)	21.94 ± 2.44	18.90 ± 2.66	<0.01	19.14 ± 3.41	15.78 ± 3.97	<0.01
ABE-V (mmo/L)	−2.80 ± 3.09	−6.67 ± 3.57	<0.01	−6.61 ± 3.86	−10.72 ± 4.60	<0.01
SBE-V (mmo/L)	−2.92 ± 3.32	−6.98 ± 3.89	<0.01	−7.15 ± 4.20	−11.55 ± 5.01	<0.01
sO_2_-V (%)	87.45 ± 6.93	84.21 ± 9.11	0.11	89.03 ± 6.49	88.00 ± 7.64	0.44
K^+^-V (mmo/L)	4.77 ± 0.65	5.07 ± 0.70	0.08	4.43 ± 0.66	4.52 ± 0.86	0.52
Na^+^-V (mmo/L)	139.39 ± 2.85	139.77 ± 2.78	0.59	141.06 ± 2.96	141.05 ± 3.13	0.99
Ca^2+^-V (mmo/L)	1.12 ± 0.09	1.12 ± 0.09	0.83	1.17 ± 0.11	1.15 ± 0.10	0.27
Cl^−^-V (mg/dL)	105.05 ± 3.73	106.44 ± 4.12	0.16	110.60 ± 4.70	114.46 ± 5.47	<0.01
AG-V (mmo/L)	11.58 ± 1.89	13.53 ± 2.53	<0.01	11.36 ± 2.26	10.94 ± 2.74	0.37
TSH (iuU/mL)	1.67 ± 1.92	1.71 ± 2.10	0.95	1.59 ± 1.07	2.12 ± 1.56	0.06
Total protein (g/dL)	6.18 ± 0.85	6.50 ± 0.68	0.15	6.17 ± 0.86	5.68 ± 0.94	<0.01
Albumin (g/dL)	3.47 ± 0.47	3.52 ± 0.50	0.69	3.50 ± 0.51	3.20 ± 0.64	<0.01
TC (mg/dL)	193.10 ± 60.27	155.54 ± 37.74	0.01	194.69 ± 67.63	189.10 ± 68.20	0.68
TG (mg/dL)	158.91 ± 108.45	111.81 ± 50.84	0.06	167.82 ± 82.68	143.91 ± 78.29	0.14
CRP (mg/L)	10.26 ± 14.37	20.75 ± 25.19	0.04	8.90 ± 17.24	9.02 ± 11.24	0.97
Pi (mg/dL)	5.57 ± 1.62	5.61 ± 1.58	0.93	5.26 ± 1.22	5.73 ± 1.29	<0.05
AP (UI/L)	94.94 ± 73.92	85.19 ± 51.94	0.60	73.20 ± 28.68	77.27 ± 29.16	0.48
BNP (pg/mL)	2752.62 ± 6919	904.26 ± 1305	0.39	532.02 ± 859	600.78 ± 662	0.71
Hb (g/dL)	9.95 ± 1.61	10.76 ± 1.93	0.07	10.48 ± 1.51	9.44 ± 1.18	<0.01
sCr (mg/dL)	6.46 ± 2.40	6.89 ± 2.53	0.48	5.37 ± 2.15	5.41 ± 1.43	0.90
eGFR (mL/min/1.75 m^2^)	10.32 ± 3.81	9.56 ± 5.46	0.51	12.22 ± 4.73	10.88 ± 3.46	0.09
Urea (mg/dL)	96.21 ± 33.48	113.38 ± 36.25	0.05	143.88 ± 45.28	150.69 ± 37.95	0.39
PTH (pg/mL)	427.67 ± 392.02	384.27 ± 251.59	0.69	311.69 ± 194.62	392.77 ± 259.20	0.09
p50(act)-A (mmHg)	25.46 ± 1.13	28.39 ± 1.67	<0.01	25.67 ± 1.23	28.37 ± 1.79	<0.01
p50(act)-V (mmHg)	26.26 ± 1.39	28.85 ± 1.23	<0.01	25.71 ± 1.36	28.13 ± 1.41	<0.01
Δp50(act) (mmHg)	−0.80 ± 1.96	−0.46 ± 1.91	0.49	−0.04 ± 1.97	0.24 ± 2.16	0.45
Δp50(act) (%)	−3.38 ± 7.79	−1.91 ± 6.63	0.43	−0.44 ± 7.52	0.51 ± 7.38	0.49
Mean p50(act) (mmHg)	25.86 ± 0.79	28.62 ± 1.11	<0.01	25.69 ± 0.84	28.25 ± 1.19	<0.01

Student’s *t*-test for the independent variables, *p* < 0.05 statistically significant. Abbreviations: pO_2_, oxygen partial pressure; pCO_2_, carbon dioxide partial pressure; HCO_3_^−^, bicarbonate; ABE, actual base excess; SBE, standard base excess; Na^+^, sodium; K^+^, potassium; Ca^2+^, ionized calcium; Cl^−^, chloride; AG, anion gap; TC, total cholesterol; TG, triglycerides; Pi, phosphate; AP, alkaline phosphatase; BNP, B-type natriuretic peptide, Hb, hemoglobin; sCr, serum creatinine; CRP, C-reactive protein; PTH, parathormone; TSH, thyroid-stimulating hormone.

## Data Availability

The data provided in this study are accessible upon request from the corresponding author.

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
