# Peer review of "Changes of Dissociative Properties of Hemoglobin in Patients with Chronic Kidney Disease"

_diagnostics, 2024, doi:10.3390/diagnostics14121219_

Round 1

Reviewer 1 Report

Comments and Suggestions for Authors

I considered the manuscript entitled “Changes of dissociative properties of hemoglobin in patients with chronic kidney disease” by Justyna Korus, et al, which is intended to be published in Diagnostics journal.

This manuscript deals with the differences on the hemoglobin dissociation curve (ODC) in two populations of CKD patients which differ between themselves on whether they are on dialysis or not. The manuscript is well written and exposed, no concerns. However, to me it does not introduce any special novelty. Authors comment at the beginning of discussion: The majority of scientific knowledge, regarding the characteristics of the ODC in individuals with CKD, is derived from physiological research carried out during the 1970s, which focused on patients with advanced CKD or those undergoing renal replacement therapy (RRT).

It is well known that metabolic acidosis moves the ODC, and CKD patients are a good example of the effect of acidosis. Patients on dialysis are correctly tamponade by bicarbonate infusion from dialysate and removal of acid compounds. Then the dissociation curve restores. Apart the effect of erythropoietin may be marginal, as its main effect is to adjust externally the level of endogenous hemoglobin.

Minor concerns

2) 108 CKD stage G4 and G5 qualified for AVF creation but not being on hemodialysis program. I do not understand this item as you then have non-HD, N=115, and HD, N=65

What do you mean with: cross-sectional study?

I did not find how you measure p50, or it is not apparent

Author Response

Dear reviewer,

On behalf of all the authors who contributed to this article, I would like to express our sincere appreciation for your time and constructive comments on our article. We have made extensive modifications to make our results convincing. In this revised version, changes to our manuscript were all highlighted in red diagonal font. Point - by – point responses to the editor and reviewer are listed below this letter.

I considered the manuscript entitled “Changes of dissociative properties of hemoglobin in patients with chronic kidney disease” by Justyna Korus, et al, which is intended to be published in Diagnostics journal.

This manuscript deals with the differences on the hemoglobin dissociation curve (ODC) in two populations of CKD patients which differ between themselves on whether they are on dialysis or not. The manuscript is well written and exposed, no concerns. However, to me it does not introduce any special novelty. Authors comment at the beginning of discussion: The majority of scientific knowledge, regarding the characteristics of the ODC in individuals with CKD, is derived from physiological research carried out during the 1970s, which focused on patients with advanced CKD or those undergoing renal replacement therapy (RRT).

It is well known that metabolic acidosis moves the ODC, and CKD patients are a good example of the effect of acidosis. Patients on dialysis are correctly tamponade by bicarbonate infusion from dialysate and removal of acid compounds. Then the dissociation curve restores. Apart the effect of erythropoietin may be marginal, as its main effect is to adjust externally the level of endogenous hemoglobin.

Response:  Thank you for your valuable feedback on our manuscript. Of course, we agree that the pathophysiology of ODC has already been known and the impact of anemia and acidosis on ODC is well described. Most of the older studies from the 1970s, i.e. in the era before the introduction of erythropoietin into therapy, and conditions of severe anemia, well indicated the importance of ODC in alleviating the symptoms of reduced oxygen supply. Currently, however, the problem of anemia in CKD patients is limited and, as the reviewer notes, the effect of ODC has a minor impact on the symptoms of tissue ischemia, because most of these patients do not develop severe anemia. However, our research demonstrated that dialysis treatment alters the dissociation curve's properties. This finding is mostly of scientific rather than practical significance.

Minor concerns

 2) 108 CKD stage G4 and G5 qualified for AVF creation but not being on hemodialysis program. I do not understand this item as you then have non-HD, N=115, and HD, N=65

Response: We appreciate that you discovered this accidental error and have therefore removed the “but not being on hemodialysis program” part.

 What do you mean with: cross-sectional study?

Response: Cross-sectional studies are observational studies that analyze data from a population at a single point in time. There is not possible to observed changes over the time.

I did not find how you measure p50, or it is not apparent.

Response: Thank you for this remark. p50 is defined as the partial pressure of oxygen (pO2) in blood at 50 % oxygen saturation (sO2). It is an estimated parameter derived from pO2 and sO2 via extrapolation along the oxyhemoglobin dissociation curve. (ODC). (Robert W Burnett. Minimizing error in the determination of P50. Clin Chem. 2002 Mar;48(3):567-70.)

We have added this description to the Methods section.

Thank you for all of your comments that helped us improve our manuscript.

Reviewer 2 Report

Comments and Suggestions for Authors

Author Response

Dear reviewer,

On behalf of all the authors who contributed to this article, I would like to express our sincere appreciation for your time and constructive comments on our article. We have made extensive modifications to make our results convincing. In this revised version, changes to our manuscript were all highlighted in red diagonal font. Point - by – point responses to the editor and reviewer are listed below this letter.

Review Diagnostics-2997383

The authors tackle an always very interesting topic. Anemia in patients with CKD has been the subject of study in recent years and research has produced many papers on this topic and on iron requirements. Patients on hemodialysis and non-dialysis CKD stages G4 and G5 were enrolled However, the paper contains important biases. In Tab 2 under the heading "Parameters of erythropoiesis" ferritin and the weekly dose of EPO have a standard deviation greater than the basal value!. The ferritin value is very high. These data (see Dopps study, Pivotal study by Mc Dougall, Lacquaniti et al KRCP 2020, Lacquaniti et al Medicina 2023, Minutolo R et al CKJ 2022) correlate with patients' inflammation. The use of ferrous carboxymaltose, which may have an anti-inflammatory effect on hepcidin expression, is not reported. These are very contradictory data. These contradictory data are not clarified in the text. The authors must explain these data in the text under discussion also based on the recommended references. There are no clinical data in the text to support the results. Authors must make these changes. In this version the paper cannot be published

Response: Thank you for your insightful comments on our manuscript.

Anaemia in chronic renal disease patients can be caused by diabetes, high blood phosphate levels, low eGFR (<30 ml/min/1.73 m2), proteinuria (>0.50 g/day), and other conditions. (Roberto Minutolo at. Al. New-onset anemia and associated risk of ESKD and death in non-dialysis CKD patients: a multicohort observational study. Clin Kidney J. 2022 Jan 12;15(6):1120-1128. doi: 10.1093/ckj/sfac004. eCollection 2022 Jun. DOI: 10.1093/ckj/sfac004). Not all predictors were deeply investigated in this study and this are limitation of the study. The reviewer rightly noted few important issues related to the correction of anemia, i.e.

  1. High standard deviation (SD) for the EPO dose.

High SD predominantly affects non-HD patients and is caused by the fact that not all patients in the non-HD group received EPO and their dosage was tailored to the recommended Hb level of 11g/dl. (KDIGO Clinical Practice Guideline for. Anemia in Chronic Kidney Disease. Kidney inter., Suppl. 2012; 2: 279–335. 282. Kidney International.). The wide range of EPO dosages is not only a matter of individualization but, as the DOPPS study shows, there is also great variation across countries from 5,200 UI/week in Japan to 17,300 UI/week in the United States. (Pisoni et al. Anemia management and outcomes from 12 countries in the Dialysis Outcomes and Practice Patterns Stud (DOPPS)Am J Kidney Dis. 2004 Jul;44(1):94-111. doi: 10.1053/j.ajkd.2004.03.023.).

  1. High ferritin concentrations and high SD.

The study examined non-HD and HD populations without evaluating a wide range of inflammatory markers, whereas none of the patients presented with active infection at the time of AVF creation. We do not estimate hepcidin level, which regulates iron metabolism by inhibiting intestinal iron absorption and iron release from iron stores and is the marker of inflammation. The DOPPS study showed that a higher Hb level above 11g/dl is associated with a lower ferritin level in dialysis patients. (Pisoni et al. Anemia management and outcomes from 12 countries in the Dialysis Outcomes and Practice Patterns Stud (DOPPS)Am J Kidney Dis. 2004 Jul;44(1):94-111. doi: 10.1053/j.ajkd.2004.03.023.). This inverse relationship may be related to hidden inflammation, e.g. in patients with catheters or non-functioning vascular grafts, as precisely described by Nassar el al. (Nassar GM, et al. Occult infection of old nonfunctioning arteriovenous grafts: A novel cause of erythropoietin resistance and chronic inflammation in hemodialysis patients. Kidney Int Suppl 80:S49-S54, 2002). Due to the reviewer's comment regarding inflammation and high ferritin levels, the results of CRP and ferritin concentrations were once more time reviewed in the database and analyzed by statistician. In around 10% of the results, we discovered an inaccuracy in which the ferritin level was entered from other hospital stays, primarily due to infectious problems, rather than the period of fistula creation (and POCT sampling). We apologize for lack of scrutiny. In the data, we analysed the CRP level, which only changed slightly and we still observed significantly higher CRP in dialysis patients. CRP level before reanalysis was (8.96±14.77 v. 14.52±20.00, p<0.05) and after reanalysis (8.78±14.77 v. 14.43±19.9 p<0.05) for non-HD and HD patients, respectively. Meanwhile, reanalysis showed no significant difference in ferritin levels between HD and non-HD patients (238.94±245.46 vs. 274.31±186.4, p>0.05) which was corrected. These ferritin levels do not exceed the recommended level of 500 ng/ml. (KDIGO Clinical Practice Guideline for. Anemia in Chronic Kidney Disease. Kidney inter., Suppl. 2012; 2: 279–335. 282. Kidney International.)

  1. Iron supplementation plays pivotal role in anemia correction. Many studies indicate that an optimum iron-deficit therapy regimen may achieve and maintain Hb levels quickly, and delay or eliminate the requirement of additional anaemia therapies, including erythropoiesis-stimulating agents (ESAs). (Macdougall IC, et al. FIND-CKD: a randomized trial of intravenous ferric carboxymaltose versus oral iron in patients with chronic kidney disease and iron deficiency anaemia. Nephrol Dial Transplant. 2014 Nov;29(11):2075-84. doi: 10.1093/ndt/gfu201.) Our study used two different iron supplementation methods, oral splementation for non-HD and intravenous for HD group, so comparison of HD with non-HD is difficult in this respect. In the HD group, the mean weekly dose of intravenously injected iron(III)-hydroxide sucrose complex was 27.46±66mg. In the non-HD group iron(III) hydroxide with polymaltose or iron(II) sulfate was used and the mean daily dose of iron of was 42.46±64.48mg. Although our analysis showed that the percentage of dialysis patients was more likely to receive iron supplementation (64% v. 35%, p>0.05), comparing the HD and non-HD groups in this respect is difficult and may also affect the correction of anemia.

We have added this information to the Results and Discussion section. 

Thank you for all of your comments that helped us improve our manuscript.

Round 2

Reviewer 1 Report

Comments and Suggestions for Authors

The authors gives a comprehensive writing concerning my comments but do not show which is the real novelty

Reviewer 2 Report

Comments and Suggestions for Authors

The authors made the requested changes